# Physical Activity, Mental Health and Wellbeing of Adults within and during the Easing of COVID-19 Restrictions, in the United Kingdom and New Zealand

**DOI:** 10.3390/ijerph19031792

**Published:** 2022-02-04

**Authors:** James Faulkner, Wendy J. O’Brien, Beth Stuart, Lee Stoner, John Batten, Daniel Wadsworth, Christopher D. Askew, Claire E. Badenhorst, Erin Byrd, Nick Draper, Catherine Elliot, Simon Fryer, Michael J. Hamlin, John R. Jakeman, Kelly A. Mackintosh, Melitta A. McNarry, Andrew Mitchelmore, Helen Ryan-Stewart, Zoe Saynor, Mia A. Schaumberg, Emily Spiegelhalter, Keeron Stone, Danielle Lambrick

**Affiliations:** 1School of Sport, Health and Community, University of Winchester, Winchester SO22 4NR, UK; John.Batten@winchester.ac.uk (J.B.); helen.ryan-stewart@winchester.ac.uk (H.R.-S.); milyspiegel@gmail.com (E.S.); 2School of Sport, Exercise and Nutrition, Massey University, Auckland 0745, New Zealand; w.j.obrien@massey.ac.nz (W.J.O.); c.badenhorst@massey.ac.nz (C.E.B.); 3Southampton Clinical Trials Unit, Faculty of Medicine, University of Southampton, Southampton SO17 1BJ, UK; bls1@soton.ac.uk; 4Department of Exercise and Sport Science, University of North Carolina, Chapel Hill, NC 27514, USA; dr.l.stoner@gmail.com; 5School of Nursing, Midwifery and Paramedicine, University of the Sunshine Coast, Queensland 4558, Australia; dwadswor@usc.edu.au; 6Sunshine Coast Health Institute, Sunshine Coast Hospital and Health Service, Queensland 4575, Australia; caskew@usc.edu.au (C.D.A.); mschaum1@usc.edu.au (M.A.S.); 7School of Health and Behavioural Sciences, University of the Sunshine Coast, Queensland 4558, Australia; 8Department of Sport, Health Sciences and Social Work, Oxford Brookes University, Oxford OX3 0BP, UK; ebyrd@brookes.ac.uk (E.B.); jjakeman@brookes.ac.uk (J.R.J.); amitchelmore@brookes.ac.uk (A.M.); 9Faculty of Health, University of Canterbury, Christchurch 4800, New Zealand; nick.draper@canterbury.ac.nz; 10Department of Tourism, Sport and Society, Lincoln University, Lincoln 7647, New Zealand; catherine.elliot@lincoln.ac.nz (C.E.); michael.hamlin@lincoln.ac.nz (M.J.H.); 11School of Sport and Exercise, University of Gloucestershire, Gloucester GL2 9HW, UK; sfryer@glos.ac.uk (S.F.); kstone1@glos.ac.uk (K.S.); 12Applied Sports, Technology, Exercise and Medicine Research Centre, Swansea University, Swansea SA1 8EN, UK; k.mackintosh@swansea.ac.uk (K.A.M.); M.Mcnarry@Swansea.ac.uk (M.A.M.); 13Physical Activity, Health and Rehabilitation Thematic Research Group, School of Sport, Health and Exercise Science, University of Portsmouth, Portsmouth PO1 2UP, UK; zoe.saynor@port.ac.uk; 14School of Human Movement and Nutrition Sciences, The University of Queensland, Queensland 4067, Australia; 15School of Health Sciences, University of Southampton, Southampton SO17 1BJ, UK; D.M.Lambrick@soton.ac.uk

**Keywords:** Coronavirus disease, pandemic, lifestyle behaviour change, exercise, depression, lockdown

## Abstract

Physical activity (PA) participation was substantially reduced at the start of the COVID-19 pandemic. The purpose of this study was to assess the association between PA, mental health, and wellbeing during and following the easing of COVID-19 restrictions in the United Kingdom (UK) and New Zealand (NZ). In this study, 3363 adults completed online surveys within 2–6 weeks of initial COVID-19 restrictions (April/May 2020) and once restrictions to human movement had been eased. Outcome measures included the International Physical Activity Questionnaire Short-Form, Depression Anxiety and Stress Scale-9 (mental health) and World Health Organisation-5 Wellbeing Index. There were no differences in PA, mental health or wellbeing between timepoints (*p* > 0.05). Individuals engaging in moderate or high volume of PA had significantly better mental health (−1.1 and −1.7 units, respectively) and wellbeing (11.4 and 18.6 units, respectively) than individuals who engaged in low PA (*p* < 0.001). Mental health was better once COVID-19 restrictions were eased (*p* < 0.001). NZ had better mental health and wellbeing than the UK (*p* < 0.001). Participation in moderate-to-high volumes of PA was associated with better mental health and wellbeing, both during and following periods of COVID-19 containment, compared to participation in low volumes of PA. Where applicable, during the current or future pandemic(s), moderate-to-high volumes of PA should be encouraged.

## 1. Introduction

The ongoing pandemic of coronavirus disease 2019 (COVID-19) has affected millions of people worldwide, causing mild to severe respiratory illness and death. Different international approaches have been implemented to control, manage and recover from the virus, with governments and their citizens still awaiting a “new normal” to emerge.

There is evidence that physical activity (PA) has strong relevance for limiting and protecting human health during the pandemic [1]. However, physical distancing and self-isolation directives, implemented by many national governments to reduce the risk of person-to-person transmission of COVID-19, have been associated with decreased PA engagement [2,3]. During the initial COVID-19 containment (lockdown) period, PA decreased by more than 25% [3,4,5,6,7], whilst physical inactivity was associated with a higher risk for severe COVID-19 outcomes, including hospitalisations, admission to intensive care units and death [8]. The University College London COVID-19 study demonstrated that a substantial proportion of their 35,915 adult sample showed persistent physical inactivity or decreasing PA during the first 22 weeks of the pandemic [3]. Furthermore, research during the initial COVID-19 containment period has consistently demonstrated an association between reduced PA and poorer mental health [2,4,9]. Given that COVID-19 will be a part of our foreseeable futures, and that we cannot discount the occurrence of future infectious disease pandemics [10], it is prudent that we better understand the impact that restrictions on human movement has on PA patterns and the associations with mental health and wellbeing.

A natural experimental approach to investigate the impact of COVID-19 containment strategies on PA and health outcomes is through cross-national comparison. For example, during the initial containment period, better mental health was reported in countries where less restrictions were imposed on factors such as location, timing and frequency of PA (e.g., New Zealand [NZ]) compared to countries where PA was limited to a single period of up to 60 min per day (e.g., United Kingdom [UK]) [2]. To date, a direct, longitudinal cross-national comparison has not been reported. Longitudinal analysis will permit more robust understanding of the relationship between PA with mental health and wellbeing as the pandemic evolves, elucidating whether these relationships are moderated over time, while the cross-national comparison will provide important insight as to the importance of containment strategies, including differences in imposed restrictions to human movement and the subsequent easing of these.

The current study recruited individuals living in the UK and NZ and collected measurements at two time points: during initial COVID-19 containment (T1) and once initial restrictions were eased (T2). The specific aims of this study were to determine: (Aim 1) the effect of easing COVID-19 restrictions on PA, mental health and wellbeing; (Aim 2) the association between PA (low, moderate or high volumes) with mental health and well-being; (Aim 3) whether the association is moderated by time (T1 vs. T2); and (Aim 4) whether the association is moderated by country (UK vs. NZ).

## 2. Materials and Methods

This study was approved by institutional human ethics committees in the UK (Approval number: HWB/REC/20/03) and NZ (Approval number: 4000022445) and is reported in accordance with the Strengthening the Reporting of Observational Studies in Epidemiology (STROBE) guidelines [11].

English speaking adults (≥18 years) who were residing in the UK or NZ and who had online access to complete the surveys were eligible to participate in the study. Convenience sampling, via mass emailing through collaborating author networks, social media (e.g., Twitter, Facebook) and mass media engagement (e.g., radio, newspapers) were used to recruit participants at T1. All participants provided informed consent at the start of T1, and on completion were asked whether they would consent to being contacted to take part in T2.

This study was conducted using two online surveys: JISC (Bristol, UK) for UK respondents, and Qualtrics (London, UK) for NZ respondents. The online survey included questions pertaining to mental health, wellbeing, PA amount and duration, and exercise intention behaviours (Stages of Change scale) [12]. Additionally, participants were asked to denote their age, sex, ethnicity, and presence of long-term health conditions (LTC). Survey one (T1) was disseminated during early government-mandated (03/04/20–12/05/20) COVID-19 containment for both the UK and NZ. Survey two (T2) was disseminated when several restrictions on these activities had been eased (although social distancing rules remained) in both the UK (01/08/20–31/08/20) and NZ (05/06/20–15/06/20).

The International Physical Activity Questionnaire: Short Form [IPAQ-SF]) [13] was used to estimate three PA categories based on the amount (days per week), duration (minutes) and intensity (Metabolic Equivalent for Task [MET]) of weekly PA: walking (3.3 MET), moderate-intensity PA (4.0 MET), and vigorous-intensity PA (8.0 MET). Average daily sitting time was also obtained [14]. The IPAQ-SF is a valid (r = 0.67) and reliable tool (rho = 0.77–1.00) for assessing PA in various age groups (e.g., 18–70 years) [13]. In our study, for both T1 and T2, MET·min^–1^·week^–1^ were calculated for each PA category and then summed to derive a total PA score. Participants’ IPAQ-SF responses were also categorised into low, moderate or high volumes of PA [2] using the following criteria: *High*: Based on seven or more days of any combination of walking, moderate- or vigorous-intensity activities achieving ≥3000 MET·min^–1^·week^–1^; *Moderate*: Based on five days or more of any combination of walking, moderate- or vigorous-intensity activities achieving ≥600 to 2999 MET·min^–1^·week^–1^; *Low*: Based on achieving ˂600 MET·min^–1^·week^–1^.

Wellbeing was captured using the World Health Organisation-5 Well-being Index (WHO-5) [15], while mental health was based on the Depression Anxiety and Stress Scale-9 (DASS-9) [16]. The total DASS score (sum of depression, anxiety and stress scores) is reported, where higher scores relate to higher overall depression, anxiety, and stress.

Statistical analyses were performed using R Statistical Software. The corresponding author had full access to the data in the study and was responsible for the integrity of the data set and the data analyses. Only those respondents who completed both T1 and T2 surveys are included within the data analysis for this study. Raw data are presented as mean (standard deviations, (SD)), or frequencies (%), where appropriate, and mixed model data are presented as standardized beta (B) and precision estimate (95% confidence interval (95%CI)). Additionally, standardized betas (β) were calculated by dividing the effect β by SD and are presented to facilitate comparisons between variables. The α level was set a priori for main effects at 0.05, and for interaction effects at 0.10.

For Aim 1, a series of repeated measures analysis of variance (ANOVA) assessed outcome measures (IPAQ-SF, DASS-9, WHO-5) between T1 and T2 for the whole sample and by country of residence (UK, NZ). For Aims 2 to 4, and for each outcome (DASS-9 and WHO-5), four linear mixed effects models were specified, with random intercepts and fixed slopes. For Aim 2, PA (low, moderate, high) was regressed against the continuous DASS-9 and WHO-5 scores using separate linear mixed models (Model 1). Subsequently, Model 2 adjusted for the potential confounders, which were centred prior to specification: age (18–29, 30–39, 40–49, 50–59, 60–69, 70–79, 80+ years), sex (Male, Female), ethnicity (White, Polynesian, Indian, Asian, Black, Mixed, Other) and LTC (with or without). For Aim 3, Model 3 specified PA by Time (during initial COVID-19 restrictions (T1) vs. following restrictions (T2)) interaction, while Model 4 specified PA by Country (UK vs. NZ) interaction term. Using Model 4, Aim 2 was tested using the slope (PA) and Aims 3 and 4 using the respective interaction terms. All models were assessed by examination of the model residuals plotted against their normal scores. The assumptions of normality and homoscedasticity were assessed via visual inspection of the frequency and residual distributions, respectively. To test for multicollinearity, variance inflation factors were compared to the recommended cut-point of 10.

## 3. Results

### 3.1. Participants

Of the 7128 participants who completed T1, 5687 (80%) consented to follow-up at T2. Of those who consented, 3363 (59.1%) completed T2 and are included in the results. Participants at T1 and T2 were predominantly white females (Table 1). T2 participants were, on average, older than T1 (49.0 ± 14.8 years vs. 45.4 ± 14.8 years, respectively), with more people in the 40–69 years age range (61.9% vs. 55.4%, T2 and T1, respectively). The majority of participants (73.7%) at T2 met PA guidelines (Table 2).

### 3.2. Change in PA, Mental Health and Wellbeing

Aim 1: there were no statistical differences in PA, DASS-9, or WHO-5 scores between T1 and T2 (Table 1). However, PA was significantly lower for NZ than the UK at T2 (Mean (95% CI); −764.2 (−932.7, −595.7) MET·min^−1^·week^−1^; *p* < 0.001). There were no differences in sitting time between countries or timepoints. A large proportion of the sample (69.3%) demonstrated no change in their exercise behaviour between T1 and T2 (Table 2). Although the UK (16.2%) and NZ (16.6%) demonstrated a similar proportion of negative changes in exercise behaviour between timepoints, the UK demonstrated more positive changes than NZ (18.9% vs. 11.3%, respectively). Females, younger participants, and people with LTCs were more likely to show negative changes in exercise behaviour between T1 and T2 than males, older participants, and people without LTCs, respectively (all *p* < 0.001; Appendix A).

### 3.3. Associations between PA and Mental Health

Associations between PA and DASS-9 are presented in Table 3 and Figure 1A,B. All aims were tested using the fully adjusted Model 4. Aim 2: there was a significant association between PA and DASS-9 (*p* < 0.001). As shown in Table 3 (Model 4), the DASS-9 score is 1.1 units lower (beneficial) for moderate vs. low PA exposure (*p* < 0.001) and 1.7 units lower for high vs. low PA exposure (*p* < 0.001). Aim 3: the Time by PA interaction was not significant (Figure 1A, *p* > 0.05), though the Time effect was with DASS-9 significantly lower at T2 compared to T1 by 0.4 units (*p* < 0.001). Aim 4: the country by time by PA interaction was not significant (Figure 1B, *p* > 0.05), though a country main effect was observed with NZ DASS-9 scores being significantly lower compared to the UK by 1.1 units (*p* < 0.001).

### 3.4. Associations between PA and Wellbeing

Associations between PA and WHO-5 are presented in Table 3 and Figure 1C,D. Model 4 is adjusted for potential confounders and includes the time, country and respective interactions terms. Aim 2: there was a significant association between PA and WHO-5 (*p* < 0.001). As shown in Table 3, the WHO-5 score is 11.4 units higher (beneficial) for moderate vs. low PA exposure (*p* < 0.001) and 18.6 units higher for high vs. low PA exposure (*p* < 0.001). Aim 3: the time interaction term was not significant (Figure 1C), nor was the main effect of time (both *p* > 0.05). Aim 4: the Country by PA interaction was significant (Figure 1D, *p* = 0.079). Compared to the UK, NZ was 5.7, 5.0, and 2.9 units higher for low, moderate and high volumes of PA, respectively (all *p* < 0.001).

## 4. Discussion

This study demonstrated no changes in PA, mental health, or wellbeing between the early COVID-19 containment period (T1) and when those restrictions had been eased (T2) (Aim 1). Individuals who engaged in more moderate or high PA had significantly better mental health and wellbeing than individuals who engaged in low PA (Aim 2). Mental health was significantly better once restrictions to human movement were eased (Aim 3), whilst individuals living in NZ exhibited better mental health and wellbeing compared to the UK (Aim 4). These findings have important implications for policy and guideline recommendations to encourage people to engage in moderate to high PA, and thus promote better mental health and wellbeing, throughout the COVID-19 pandemic and recovery period.

There were no statistical differences in total PA between T1 and T2 (Table 2), despite PA reducing on average by 7%. This finding may demonstrate that the negative effects of COVID-19 containment on PA continue to exist when restrictions are eased. However, this finding appears to be driven by the NZ data, wherein at T2, NZ participants reported statistically lower levels of total PA compared to UK participants (−23.5%; Table 2). It is plausible that the seasons in which the T2 surveys were administered (UK summer; NZ winter) may have influenced this, as PA has been shown to vary with seasonality [17]. Poor or extreme weather is often associated with the winter season (e.g., increased precipitation, reduced daylight), and has been identified as an important barrier to participation in PA, especially outdoors [17]. Accordingly, UK participants demonstrated more positive changes in exercise behaviour between T1 and T2 compared to NZ (Table 2). With surges in COVID-19 cases already associated with winter seasons [18], future policies and guidance regarding PA interventions should consider how seasonality effects PA behaviours. Facilitating COVID-safe indoor opportunities and providing guidance as to how to be physically active during the COVID-19 pandemic is paramount.

Individuals who engaged in a high volume of PA were 1.7 units more likely to exhibit better mental health, and 18.6 units more likely to demonstrate better wellbeing compared to people who engaged in low PA (Table 3). These findings support previous research that investigated similar outcomes during the initial COVID-19 lockdown that PA can help protect human health during the pandemic [2,4,19]. Although the magnitude of difference was smaller, favourable findings were also observed when comparing moderate PA with low PA for both mental health (−1.1 units) and wellbeing (11.4 units better). As COVID-19 infection ‘waves’ have resulted in continued implementation and subsequent easing of restrictions on human movement through government-imposed lockdowns and mandatory isolation, developing appropriate and tailored support mechanisms to facilitate individual’s (re)engagement in moderate or high PA is essential for optimising people’s mental health and wellbeing, especially during times of heightened anxiety and stress such as stages of pandemic restriction.

Despite no changes in wellbeing, mental health was significantly better at T2 compared to T1 (−0.4 units). Research from the early COVID-19 containment period typically demonstrated an association between reduced PA and poorer mental health [2,4,9]. Individuals who did not meet PA guidelines and/or engaged in more screen time during the initial COVID-19 containment reported higher levels of depression and stress than those who exercised more during this period [4]. Our study clearly provides evidence to advocate a strong public health message of engaging in moderate-to-high PA during periods of COVID-19 containment, and beyond, to support individuals’ mental health. Although social distancing will continue to be important, providing people with greater opportunities to engage in PA, such as those seen in T2, is an important and effective strategy to protect people and healthcare systems from exacerbating the increasing prevalence of mental health issues that has manifested from the pandemic [20].

Our study demonstrated a stronger association between PA and mental health (−1.1 units), and between all volumes of PA (low, moderate, high) and wellbeing, in NZ compared to UK respondents. This finding supports our earlier study whereby NZ participants demonstrated better mental health and wellbeing during the initial COVID-19 containment period than UK, Ireland, and Australian participants [2]. Furthermore, previous research during the initial COVID-19 containment period has shown that, when adjusted for age and gender differences, anxiety and stress were significantly lower in NZ than in the UK [21]. It is plausible that the significantly higher cases of COVID-19 and related mortalities reported in the UK than NZ may have resulted in our UK sample feeling more at risk and anxious of being infected [21]. Similarly, different restrictions in social distancing and containment efforts across both countries, as well as the UK having a much larger population, may have made the UK sample feel less protected and more exposed to the virus [21], which may have influenced feelings of wellbeing across the two countries. Furthermore, NZ has substantial green space [22], and although PA was lower in NZ than the UK at T2, PA in greenspaces has been shown to be better for mental health and wellbeing than PA conducted in urban environments [23,24]. As these factors were not measured in our study, future research should explore in greater depth the underlying reasons for differences in mental health and wellbeing between the UK and NZ.

When considering the limitations to the study, and as detailed elsewhere [2], white females were the predominant respondents to our survey. Further investigation into the relationship between PA and mental health during COVID-19 in various ethnic and cultural groups is needed when considering that racial and ethnic disparities may impact the burden of COVID-19 related outcomes [25]. Furthermore, we recruited an atypical active sample, whereby 73% of participants reported that they met recommended PA guidelines; higher than the population average of the countries surveyed [26,27]. The recruitment of a more heterogenous population may provide different insight into the association between PA and mental health and wellbeing, and as such warrants future research consideration. We also lack data on PA before the start of the COVID-19 pandemic, so it remains unknown how change or stability in PA observed in this study compare to usual levels in our sample before the pandemic [3]. There are many strengths to this study, including: the longitudinal study design, the large sample size, and the speed with which the T1 survey was administered. A further strength was that both T1 and T2 surveys were administered in the UK and NZ when similar restrictions to human movement were imposed.

## 5. Conclusions

Our study provides meaningful insights into people’s PA behaviours both during, and importantly following, the easing of initial COVID-19 restrictions in different countries. Despite no changes in total PA, mental health, or wellbeing from before- to after periods of COVID-19 containment, the UK demonstrated more positive changes in exercise behaviour than NZ, plausibly due to the T2 survey being disseminated in the UK summer season. Individuals who engaged in a moderate-to-high volume of PA had better mental health and wellbeing than people who engaged in less PA. After easing of COVID-19 containment, PA was more strongly associated with better mental health and wellbeing. Furthermore, people in NZ reported better mental health and wellbeing than people from the UK. With the continued risk of COVID-19, and future concern aligned with new global viruses and pandemics, the findings of this study have important implications for local-, regional-, and national policy and guideline recommendations, and may assist government strategies and future directives around the importance of promoting PA within national containment approaches.

## Figures and Tables

**Figure 1 ijerph-19-01792-f001:**
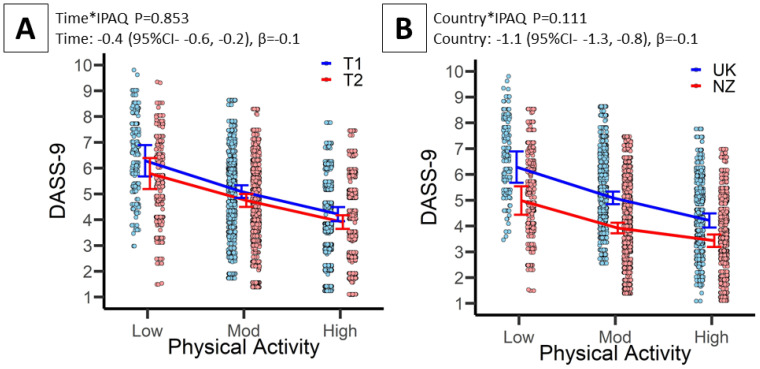
Associations between subjectively measured physical activity (IPAQ-SF) with mental health (DASS-9) and well-being (WHO-5). Data presented are linear mixed-effects model estimates, corresponding to model 4 described in the statistical analyses section. (**A**) Association between IPAQ-SF and DASS-9 specifying the time (T2-T1) by IPAQ interaction term; (**B**) association between IPAQ-SF and DASS-9 specifying the country (NZ-UK) by IPAQ interaction term; (**C**). association between IPAQ-SF and WHO-5 specifying the time (T2-T1) by IPAQ interaction term; and (**D**) association between IPAQ-SF and WHO-5 specifying the country (NZ-UK) by IPAQ interaction term. Data points = 6728 (1333 UK * 2 time points, 2030 NZ * 2 time points). *Abbreviations:* CI, Confidence interval; DASS-9, Depression, Anxiety and Stress Scale; IPAQ-SF, International Physical Activity Questionnaire-Short-Form; Mod, Moderate; NZ, New Zealand; UK, United Kingdom; T1, Timepoint 1; T2, Timepoint 2; WHO-5, World Health Organisation-5 Well-being Index.

**Table 1 ijerph-19-01792-t001:** Population characteristics by country for Survey 1 (T1) and 2 (T2).

		T1			T2	
	UK	NZ	Total	UK	NZ	Total
Sample [*n*]	3121	4007	7128	1333	2030	3363
Sex [*n* (%)]						
Male	1024 (32.8%)	1087 (27.1%)	2111 (29.7%)	426 (32.0%)	507 (25.0%)	933 (27.7%)
Female	2094 (67.1%)	2886 (72.0%)	4980 (70.0%)	905 (67.9%)	1510 (74.4%)	2415 (71.8%)
Prefer not to say	3 (0.1%)	24 (0.9%)	27 (0.4%)	2 (0.2%)	13 (0.6%)	15 (0.5%)
Mean age (years) [x(sd)]	43.9 (14.9)	46.5 (14.7)	45.4 (14.8)	48.6 (15.0)	49.3 (14.7)	49.0 (14.8)
Age groups [*n* (%)]						
20–29 years	627 (20.2%)	619 (15.5%)	1246 (17.5%)	171 (12.9%)	229 (11.3%)	400 (11.9%)
30–39 years	722 (23.2%)	775 (19.3%)	1497 (21.0%)	249 (18.7%)	339 (16.7%)	588 (17.5%)
40–49 years	650 (20.9%)	910 (22.7%)	1560 (21.9%)	260 (19.6%)	447 (22.0%)	707 (21.0%)
50–59 years	559 (18.0%)	853 (21.3%)	1412 (19.9%)	293 (22.0%)	460 (22.7%)	753 (22.4%)
60–69 years	392 (12.6%)	578 (14.4%)	970 (13.6%)	253 (19.0%)	386 (18.1%)	621 (18.5%)
70–79 years	141 (4.5%)	250 (6.2%)	391 (5.5%)	97 (7.3%)	170 (8.4%)	267 (8.0%)
80+ years	15 (0.5%)	22 (0.5%)	37 (0.5%)	7 (0.7%)	17 (0.8%)	24 (0.7%)
Ethnicity						
White	3010 (96.4%)	3630 (90.6%)	6640 (93.2%)	1295 (97.2%)	1886 (92.9%)	3181 (94.6%)
Polynesian	0 (0.0%)	156 (3.9%)	156 (2.2%)	0 (0.0%)	62 (3.1%)	62 (1.8%)
Indian	37 (1.2%)	48 (1.2%)	85 (1.2%)	14 (1.1%)	23 (1.1%)	37 (1.1%)
Asian	0 (0.0%)	95 (2.4%)	95 (1.3%)	0 (0.0%)	29 (1.4%)	29 (0.9%)
Black	14 (0.5%)	8 (0.2%)	22 (0.3%)	6 (0.5%)	2 (0.1%)	8 (0.2%)
Mixed groups	37 (1.2%)	18 (0.5%)	55 (0.8%)	13 (1.0%)	8 (0.4%)	21 (0.6%)
Other	23 (0.7%)	52 (1.3%)	75 (1.1%)	5 (0.4%)	20 (1.0%)	25 (0.7%)
Living situation [*n* (%)]						
Live alone	-	459 (11.5%)	-	179 (13.5%)	261 (12.9%)	440 (13.1%)
Couple	-	1287 (32.1%)	-	525 (39.5%)	720 (35.5%)	1245 (37.1%)
Two parent family	-	1164 (29.1%)	-	381 (28.7%)	553 (27.2%)	934 (27.8%)
Single parent family	-	114 (2.9%)	-	32 (2.4%)	59 (2.9%)	91 (2.7%)
Extended family	-	588 (14.7%)	-	149 (11.2%)	266 (13.1%)	415 (12.4%)
Shared occupancy	-	395 (9.9%)	-	30 (2.3%)	171 (8.4%)	201 (6.0%)
Residential care	-	0 (0.0%)	-	33 (2.5%)	0 (0.0%)	33 (1.0%)
Living with LTC [*n* (%)]	-	873 (21.8%)	-	346 (26.0%)	468 (24.0%)	814 (24.8%)

*Abbreviations:* LTC, Long-term conditions; NZ, New Zealand; T1, Timepoint 1; T2, Timepoint 2; UK, United Kingdom. Black includes African, Caribbean, African American; Indian includes East Indian, Pakistani, Bangladeshi; Asian includes Chinese, Japanese and all South-East Asian; Polynesian includes Māori, Pacific Islanders.

**Table 2 ijerph-19-01792-t002:** Physical activity (IPAQ-SF), mental health (DASS-9) and wellbeing (WHO-5) for T1 and T2 for UK, NZ and total respondents. Data presented as n (%) or Mean (SD).

		T1			T2	
	UK	NZ	Total	UK	NZ	Total
Met PA guidelines [*n* (%)]				983 (73.9)	1494 (73.6)	2477 (73.7)
Stages of change [*n* (%)]						
Precontemplation	17 (1.3)	9 (0.4)	26 (0.8)	13 (1.0)	7 (0.3)	20 (0.6)
Contemplation	56 (4.2)	57 (2.8)	113 (3.4)	63 (4.7)	95 (4.7)	158 (4.7)
Preparation	216 (16.2)	266 (13.1)	482 (14.3)	238 (17.9)	349 (17.2)	587 (17.5)
Action	220 (16.5)	340 (16.8)	560 (16.7)	127 (9.5)	218 (10.7)	345 (10.3)
Maintenance	824 (61.8)	1358 (66.9)	2182 (64.9)	890 (66.9)	1361 (67.0)	2251 (67.0)
IPAQ-SF [x (SD)]						
Total PA (MET·min^–1^·week^–1^)	3109 (2322)	2979 (2219)	3030 (2261)	3302 (2545)	2527 * (2323)	2833 (2443)
Sitting time (min)	458 (263)	455 (171)	456 (212)	414 (255)	429 (193)	423 (220)
IPAQ-SF [*n* (%)]						
Low	95 (7.1)	127 (6.3)	222 (6.6)	99 (7.4)	307 (15.1)	406 (12.1)
Moderate	693 (52.0)	1125 (55.4)	1818 (54.1)	623 (46.8)	1134 (55.9)	1757 (52.3)
High	545 (40.9)	778 (38.3)	1323 (39.3)	610 (45.8)	589 (29.0)	1199 (35.7)
Exercise Behaviour (T2-T1) [n (%)]						
Positive change				251 (18.9)	229 (11.3)	480 (14.3)
Negative change				215 (16.2)	337 (16.6)	552 (16.4)
No change				865 (65.0)	1464 (72.1)	2329 (69.3)
WHO-5 [x (sd)]	55.6 (21.5)	59.9 (20.7)	58.2 (21.1)	54.8 (22.6)	57.8 (21.1)	56.6 (21.7)
DASS-9 [x (sd)]						
Depression	2.24 (2.02)	1.89 (1.76)	2.03 (1.88)	2.03 (2.08)	1.61 (1.72)	1.78 (1.88)
Anxiety	0.71 (1.39)	0.46 (1.04)	0.56 (1.20)	0.73 (1.43)	0.57 (1.20)	0.63 (1.30)
Stress	2.25 (1.94)	1.80 (1.65)	1.98 (1.78)	2.21 (2.08)	1.67 (1.63)	1.88 (1.84)
Total	5.20 (4.52)	4.15 (3.65)	4.57 (4.05)	4.97 (4.79)	3.85 (3.81)	4.29 (4.26)

*Abbreviations:* DASS, Depression, Anxiety and Stress Scale; IPAQ-SF, International Physical Activity Questionnaire; NZ, New Zealand; PA, Physical activity; T1, Survey 1; T2, Survey 2; UK, United Kingdom; WHO-5, World Health Organisation-5 Well-being Index. * Significant difference in total PA between UK and NZ at T2 (*p* < 0.001).

**Table 3 ijerph-19-01792-t003:** Associations between subjectively measured physical activity (IPAQ-SF) with mental health (DASS-9) and well-being (WHO-5). Data presented are linear mixed-effects model estimates. Specifically, simple effects for low vs. moderate- and low vs. high-physical activity. Data points = 6726 (1333 UK * 2 time points, 2030 NZ * 2 time points).

	DASS-9					WHO-5				
	B	LCI	UCI	β	*p*	B	LCI	UCI	β	*p*
Model 1. Unadjusted										
Mod−Low	−1.1	−1.4	−0.8	−0.1	<0.001	12.0	10.3	13.7	0.2	<0.001
High−Low	−1.8	−2.2	−1.5	−0.1	<0.001	19.6	17.8	21.4	0.3	<0.001
Model 2. Adjusted for age, sex & ethnicity										
Mod−Low	−0.9	−1.2	−0.6	−0.1	<0.001	10.8	9.2	12.4	0.2	<0.001
High−Low	−1.4	−1.7	−1.1	−0.1	<0.001	17.5	15.8	19.2	0.2	<0.001
Model 3. Addition of PA by Time interaction term										
Mod−Low	−1.0	−1.3	−0.7	−0.1	<0.001	11.2	9.5	12.8	0.2	<0.001
High−Low	−1.5	−1.9	−1.2	−0.1	<0.001	17.8	16.0	19.5	0.2	<0.001
Model 4. Addition of PA by Country interaction term										
Mod−Low	−1.1	−1.4	−0.7	−0.1	<0.001	11.4	9.7	13.2	0.2	<0.001
High−Low	−1.7	−2.1	−1.4	−0.1	<0.001	18.6	16.7	20.4	0.2	<0.001

*Abbreviations:* DASS-9, Depression, Anxiety and Stress Scale; PA, physical activity; WHO-5, World Health Organisation-5 Well-being Index.

## Data Availability

The datasets used and/or analysed during the current study are available from the corresponding author on reasonable request.

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
