# Peer review of "Physical Activity, Mental Health and Wellbeing of Adults within and during the Easing of COVID-19 Restrictions, in the United Kingdom and New Zealand"

_ijerph, 2022, doi:10.3390/ijerph19031792_

Round 1
Reviewer 1 Report
This is a useful study conducted at a sensitive time that may influence national/international physical activity policy, even outside of pandemics, to ensure that physical activity environments are maintained or even enhanced. The study has been rationalised well, the method conducted appropriately and the data analysed objectively to address clear aims. The study explained its generally cross-sectional data well in the discussion.
Minor suggestions
Be consistent with use of PAV. I'd prefer to see old faithful (PA) used constantly but if you are introducing PAV as a modified concept, use it throughout and perhaps consider defining why the V is a useful addition in the introduction, ie. higher levels of PA associated with good mental health not merely avoiding being sedentary.
The sentence in line 160 is incomplete, perhaps going missing when the tables were inserted.
Reference 20 is an important modifier/moderator for this study and has been used in a diplomatic manner.
Reviewer 2 Report
This is a well written and interesting work on the effects of physical activity (PA) on mental health and wellbeing of adults within and during the easing of COVID-19 restrictions in UK and New Zeland.
It could be accepted after the following revisions.
-A brief introduction on the current pandemic of COVID-19 would be useful at the very beginning of the introduction
-after ref. 1,2 the authors should mention that PA can protect human heatlh against COVID-19
The same mention can be done after ref 18 in Discussion (line 242)
-line 102: meaning of the acronym MET should be explained at his first usage: Metabolic equivalent of task?
-tab1: why the living situation was not reported for UK? Was not asked?
-lines 217-218: 'Mental health was significantly lower (better) ' Can not the adjective 'lower' be avoided?
-line 233: ref 16 should be reported as [16]
-Both anxiety and stress were lower in NZ than UK during the initial COVID-19 containment period. Could the different feelings of people be determined also by the different environmental conditions? More people feeling at risk in polluted and overpopulated areas in UK?
-line 281: consider to place limitations in a separated section
-Supplementary Materials: do not the changes in exercise behaviours reported in Tab S1 distinguish between UK and NZ?
